# Risk of Complications in Patients Undergoing Completion Thyroidectomy after Hemithyroidectomy for Thyroid Nodule with Indeterminate Cytology: An Italian Multicentre Retrospective Study

**DOI:** 10.3390/cancers14102472

**Published:** 2022-05-17

**Authors:** Gian Luigi Canu, Fabio Medas, Federico Cappellacci, Alessio Biagio Filippo Giordano, Angela Gurrado, Claudio Gambardella, Giovanni Docimo, Francesco Feroci, Giovanni Conzo, Mario Testini, Pietro Giorgio Calò

**Affiliations:** 1Department of Surgical Sciences, University of Cagliari, 09042 Monserrato, CA, Italy; fabiomedas@unica.it (F.M.); f.cappellacci@studenti.unica.it (F.C.); pgcalo@unica.it (P.G.C.); 2Department of General and Oncologic Surgery, Santo Stefano Hospital, 59100 Prato, PO, Italy; alessiobiagio.giordano@uslcentro.toscana.it (A.B.F.G.); francesco.feroci@uslcentro.toscana.it (F.F.); 3Department of Biomedical Sciences and Human Oncology, University of Bari “Aldo Moro”, 70124 Bari, BA, Italy; angela.gurrado@uniba.it (A.G.); mario.testini@uniba.it (M.T.); 4Department of Medical and Advanced Surgical Sciences, University of Campania “Luigi Vanvitelli”, 80131 Naples, NA, Italy; claudio.gambardella2@unicampania.it (C.G.); giovanni.docimo@unicampania.it (G.D.); 5Department of Traslational Medical Sciences, University of Campania “Luigi Vanvitelli”, 80131 Naples, NA, Italy; giovanni.conzo@unicampania.it

**Keywords:** indeterminate thyroid nodule, differentiated thyroid carcinoma, total thyroidectomy, hemithyroidectomy, completion thyroidectomy, complications

## Abstract

**Simple Summary:**

The increasing use of high-quality imaging techniques together with improved access to healthcare has led to an increase in the detection of thyroid nodules. Fine-needle aspiration cytology (FNAC) is currently considered the most accurate examination for the assessment of thyroid nodular disease. However, in about 25% of cases, FNAC leads to the diagnosis of an indeterminate thyroid nodule, which represents a problem because malignancy, although relatively low (up to 30%), cannot be excluded with certainty. According to the 2015 American Thyroid Association guidelines, patients with thyroid nodular disease with an indeterminate cytology can undergo, based on established factors, a total thyroidectomy or a hemithyroidectomy. However, if an intermediate or high-risk differentiated thyroid carcinoma is detected after the hemithyroidectomy, through histological examination, the above-mentioned guidelines recommend performing a completion thyroidectomy. The main aim of this study was to assess the rate of complications in patients undergoing a completion thyroidectomy after a hemithyroidectomy for a thyroid nodule with an indeterminate cytology.

**Abstract:**

There is still controversy as to whether patients undergoing a completion thyroidectomy after a hemithyroidectomy for a thyroid nodule with an indeterminate cytology have a comparable, increased or decreased risk of complications compared to those submitted to primary thyroid surgery. The main aim of this study was to investigate this topic. Patients undergoing a thyroidectomy for thyroid nodular disease with an indeterminate cytology in four high-volume thyroid surgery centres in Italy, between January 2017 and December 2020, were retrospectively analysed. Based on the surgical procedure performed, four groups were identified: the TT Group (total thyroidectomy), HT Group (hemithyroidectomy), CT Group (completion thyroidectomy) and HT + CT Group (hemithyroidectomy with subsequent completion thyroidectomy). A total of 751 patients were included. As for the initial surgery, 506 (67.38%) patients underwent a total thyroidectomy and 245 (32.62%) a hemithyroidectomy. Among all patients submitted to a hemithyroidectomy, 66 (26.94%) were subsequently submitted to a completion thyroidectomy. No statistically significant difference was found in terms of complications comparing both the TT Group with the HT + CT Group and the HT Group with the CT Group. The risk of complications in patients undergoing a completion thyroidectomy after a hemithyroidectomy for a thyroid nodule with an indeterminate cytology was comparable to that of patients submitted to primary thyroid surgery (both a total thyroidectomy and hemithyroidectomy).

## 1. Introduction

The increasing use of high-quality imaging techniques together with improved access to healthcare has led to an increase in the detection of thyroid nodules, which have thus become a common clinical problem. At the same time, there has been an increase in the diagnosis of thyroid carcinoma, which has increased by 300% worldwide over the last three decades. The increase in the incidence rate of thyroid cancer is almost entirely due to the increase in the diagnosis of differentiated thyroid carcinoma (DTC), whose mortality has however remained relatively low and stable [1,2,3,4].

Fine-needle aspiration cytology (FNAC) is currently considered the most accurate examination for the assessment of thyroid nodular disease, given its high specificity (92%) and sensitivity (83%). However, in about 25% of cases, the FNAC leads to the diagnosis of an indeterminate thyroid nodule, which represents a problem because malignancy, although relatively low (up to 30%), cannot be excluded with certainty. In patients with this diagnosis, based on the risk of malignancy established through preoperative investigations, surgery or surveillance may be indicated. As regards surgery, in these patients, it has a diagnostic rather than therapeutic purpose, with almost 80% of surgical procedures unnecessary [5,6,7,8,9,10,11,12,13,14,15].

According to the 2015 American Thyroid Association (ATA) guidelines, patients with thyroid nodular disease with an indeterminate cytology can undergo a total thyroidectomy or a hemithyroidectomy. The choice between these two surgical procedures is influenced by several factors, including: a nodule size greater than 4 cm, a familial history of DTC, previous radiation exposure, highly suspicious ultrasound features and positivity for known mutations specific for DTC. In addition, all these factors must be further associated with the presence of bilateral nodular disease, the possible coexistence of hyperthyroidism, the patient’s medical comorbidities and patient preference. However, if an intermediate or high-risk DTC is detected after a hemithyroidectomy, through final histological examination, the above-mentioned guidelines recommend performing a completion thyroidectomy, as a hemithyroidectomy is considered oncologically inadequate in these cases [15].

There is still controversy as to whether patients undergoing a completion thyroidectomy for this reason have a comparable, increased or decreased risk of complications compared to those submitted to primary thyroid surgery [16,17,18,19,20,21,22].

The main aim of this study was to assess the rate of complications (recurrent laryngeal nerve injury, hypoparathyroidism, cervical haematoma and wound infection) in patients undergoing a completion thyroidectomy after a hemithyroidectomy for a thyroid nodule with an indeterminate cytology.

## 2. Materials and Methods

This is a multicentre, retrospective, observational study on patients who underwent thyroidectomy for thyroid nodular disease with indeterminate cytology between January 2017 and December 2020.

Data were collected from four high-volume thyroid surgery centres in Italy:-Multi-specialty General Surgery Unit, Cagliari University Hospital, Monserrato (CA);-General, Oncologic, Mininvasive and Bariatric Surgery Unit, University Hospital “L. Vanvitelli”, Naples (NA);-Academic General Surgery Unit “V. Bonomo”, Bari University Hospital, Bari (BA);-General Surgery Unit, Prato Hospital “Santo Stefano”, Prato (PO).

All participating centres used the same dedicated Microsoft Office Excel database (Microsoft Corporation, Redmond, WA, USA) for data collection.

Only patients undergoing conventional open thyroidectomy were included in this study. Patients simultaneously submitted to parathyroidectomy, those with preoperative diagnosis of lymph node metastasis, those with suspected medullary thyroid carcinoma (due to elevated preoperative calcitonin levels) and those with incomplete data were excluded.

Demographic and preoperative data, information about the surgical procedure, postoperative stay, histological findings and complications were assessed.

Based on the surgical procedure performed, four groups were identified:-TT Group: total thyroidectomy;-HT Group: hemithyroidectomy (including all hemithyroidectomies performed as initial surgery);-CT Group: completion thyroidectomy;-HT + CT Group: hemithyroidectomy with subsequent completion thyroidectomy (considered as single bilateral surgical procedure).

The primary endpoint was to assess the occurrence of complications (recurrent laryngeal nerve injury, hypoparathyroidism, cervical haematoma and wound infection) in patients undergoing completion thyroidectomy.

For this purpose, we compared:-TT Group with HT + CT Group (evaluated as bilateral surgical procedures);-HT Group with CT Group (evaluated as unilateral surgical procedures).

The secondary endpoint was to compare HT Group with CT Group in terms of operative time.

In all enrolled patients, preoperative evaluation included: medical history, physical examination, blood tests to assess thyroid function, high-resolution ultrasound (US) of the neck and fine-needle aspiration cytology.

Hyperthyroidism status was defined in the case of low serum thyroid-stimulating hormone (TSH) (<0.4 mIU/L) and use of thyrostatic drugs.

High-resolution US of the neck was always performed before surgery by an experienced surgeon, with careful evaluation of the thyroid gland and cervical lymph nodes.

The results of fine-needle aspiration cytology were classified according to the 2014 Italian consensus of the Italian Thyroid Association (AIT), the Italian Association of Clinical Endocrinologists (AME), the Italian Society of Endocrinology (SIE) and the Italian Society for Anatomic Pathology and Cytology joint with the Italian Division of the International Academy of Pathology (SIAPEC-IAP) for the classification and reporting of thyroid cytology [23].

Moreover, all patients underwent preoperative fibrolaryngoscopy to assess vocal cord mobility.

The indication and type of initial surgery (total thyroidectomy or hemithyroidectomy) were established according to the 2015 American Thyroid Association guidelines. After final histological examination, completion thyroidectomy, with or without central neck dissection, was performed in accordance with the same guidelines [15].

Recurrent laryngeal nerves and parathyroid glands were systematically searched and identified.

Intraoperative nerve monitoring (IONM), energy-based device and drain were used according to the preference of the operating surgeon.

The duration of the surgical procedure was estimated, in minutes, from skin incision to skin closure.

Postsurgical hypoparathyroidism was defined as iPTH < 10 pg/mL following the operation (normal range = 10–65 pg/mL). Permanent hypoparathyroidism was defined as iPTH levels below the normal range for more than 12 months.

Recurrent laryngeal nerve injury was diagnosed through postoperative fibrolaryngoscopy. After surgery, fibrolaryngoscopy was performed in the case of suspected recurrent laryngeal nerve injury for loss of signal at IONM or hoarseness. Recurrent laryngeal nerve injury was defined as permanent in the case of vocal cord paralysis, identified through fibrolaryngoscopy, persisting for more than 12 months.

Statistical analyses were performed with MedCalc^®^ 20.106. Chi-squared test or Fisher exact test were used for categorical variables and t-test or Mann–Whitney U test for continuous variables. *p* values < 0.05 were considered statistically significant.

## 3. Results

A total of 751 patients met the established inclusion criteria: 242 (32.22%) males and 509 (67.78%) females, with a mean age of 51.74 ± 14.15 years.

As for the initial surgery, 506 (67.38%) underwent a total thyroidectomy (TT Group) and 245 (32.62%) a hemithyroidectomy (HT Group). Detailed data regarding features of the whole sample and of these two groups are reported in Table 1 and Table 2.

Among all patients submitted to a hemithyroidectomy, 66 (26.94%) subsequently underwent a completion thyroidectomy. In patients undergoing a hemithyroidectomy with a subsequent histological finding of malignancy, a completion thyroidectomy was performed in 56.90% of the cases. One (1.52%) patient underwent a two-stage thyroidectomy for a loss of signal at intraoperative nerve monitoring. A central neck dissection was performed in 13 (19.70%) patients. The average interval between an HT and a CT was 2.12 ± 1.20 months. Malignancy in the contralateral lobe was found in 16 (24.24%) patients, while lymph node metastasis was found in 9 (13.64%).

These results are shown in Table 3.

### 3.1. Complications in TT Group and HT + CT Group

In the TT Group, there were 141 (27.87%) cases of transient hypoparathyroidism, 12 (2.37%) cases of permanent hypoparathyroidism, 4 (0.79%) cervical haematomas, 17 (3.36%) unilateral recurrent laryngeal nerve lesions, 13 (2.57%) transient recurrent laryngeal nerve lesions, 4 (0.79%) permanent recurrent laryngeal nerve lesions and 1 (0.20%) wound infection. No bilateral recurrent laryngeal nerve injury was observed.

In the HT + CT Group, there were 14 (21.21%) cases of transient hypoparathyroidism, 2 (3.03%) cases of permanent hypoparathyroidism, 4 (6.06%) unilateral recurrent laryngeal nerve lesions, 3 (4.55%) transient recurrent laryngeal nerve lesions and 1 (1.52%) permanent recurrent laryngeal nerve lesion. No bilateral recurrent laryngeal nerve injury, cervical haematoma or wound infection was observed.

No statistically significant difference was found in terms of the occurrence of hypoparathyroidism, cervical haematoma, recurrent laryngeal nerve injury and wound infection between the two groups.

These data are reported in Table 4.

### 3.2. Comparison between HT Group and CT Group

In the HT Group, the median operative time was 50 min (IQR, 42–65 min). Intraoperative nerve monitoring, an energy-based device and a drain were used in 140 (57.14%), 245 (100%) and 73 (29.80%) patients, respectively. The average postoperative stay was 1.72 ± 0.76 days. About complications, there was one (0.41%) cervical haematoma, seven (2.86%) unilateral recurrent laryngeal nerve lesions, five (2.04%) transient recurrent laryngeal nerve lesions and two (0.82%) permanent recurrent laryngeal nerve lesions. No wound infection was observed.

In the CT Group, the median operative time was 50 min (IQR, 40–60 min). Intraoperative nerve monitoring, an energy-based device and a drain were used in 33 (50%), 64 (96.97%) and 19 (28.79%) patients, respectively. The average postoperative stay was 1.77 ± 0.46 days. About complications, there were two (3.03%) unilateral recurrent laryngeal nerve lesions, one (1.52%) transient recurrent laryngeal nerve lesion and one (1.52%) permanent recurrent laryngeal nerve lesion. No cervical haematoma or wound infection was observed.

No statistically significant difference was found in terms of the operative time, the use of intraoperative nerve monitoring, the use of drain, postoperative stay and complications between the two groups.

On the contrary, the use of the energy-based device was significantly greater in the HT Group than in the CT Group (*p* = 0.045).

These data are shown in Table 5.

## 4. Discussion

The thyroidectomy, in particular, the total thyroidectomy, is the most performed surgical procedure in endocrine surgery, for the treatment of both benign and malignant diseases [1,3,7].

Although the mortality rate due to this operation is very low, complications may occur in a considerable number of patients, even in the most experienced hands and in high-volume centres. Morbidity is mainly represented by hypoparathyroidism, recurrent laryngeal nerve injury, cervical haematoma and wound infection. These complications lead to a reduction in the quality of life of the patients and increased costs for healthcare systems [24,25,26,27,28,29,30,31,32,33].

Reoperative thyroid surgery is traditionally considered to have a higher risk of complications due to the development after the primary operation, depending on the time interval between the two procedures, of tissue inflammation, oedema, adhesions and scar tissue, which make dissection difficult and result in a distortion of the normal neck anatomy causing the loss of landmarks [34,35].

Indications for a completion thyroidectomy are mainly three: a residual or recurrent thyroid carcinoma, a residual or recurrent benign thyroid disease and the detection at the histological examination of the intermediate or high-risk DTC after a hemithyroidectomy [14,15,34,35].

In the first two categories, it is generally accepted that a completion thyroidectomy entails a higher risk of complications. Differently, as regards the third category, there is still controversy about this topic [16,17,18,19,20,21,22,34,35].

Moreover, there is still uncertainty as to whether the timing of a completion thyroidectomy in patients with detection at the final histological examination of the intermediate or high-risk DTC after a hemithyroidectomy may influence the risk of complications. Some authors recommend performing completion surgery either within the first 10 days or after 3 months, reporting increased complication rates in a completion thyroidectomy performed in the intermediate postoperative period (between 10 days and 3 months). However, other studies concluded that timing has no influence on morbidity [36,37].

The main aim of this study was to assess the complication rate in this category of patients, particularly in those undergoing a completion thyroidectomy after a hemithyroidectomy for a thyroid nodule with indeterminate cytology.

Data for this analysis were collected from four high-volume thyroid surgery centres in Italy, with consequent high experience in this field of surgery.

In order to eliminate the bias related to different surgical techniques, only patients undergoing a conventional open thyroidectomy were included, while patients submitted to a minimally invasive video-assisted thyroidectomy (MIVAT), transoral endoscopic thyroidectomy vestibular approach (TOETVA) and robot-assisted transaxillary thyroidectomy (RATT) were excluded. Moreover, in order to more accurately assess postsurgical hypoparathyroidism, patients simultaneously submitted to a parathyroidectomy were also not included.

In our analysis, no statistically significant difference was found in terms of the occurrence of hypoparathyroidism, cervical haematoma, recurrent laryngeal nerve injury and wound infection comparing both the TT Group with the HT + CT Group (evaluated as bilateral surgical procedures) and the HT Group with the CT Group (evaluated as unilateral surgical procedures).

Moreover, with regard to the secondary endpoint, which was the comparison of the HT Group with the CT Group in terms of operative time, no statistically significant difference was found. In this regard, it is important to underline that operative times were comparable even though the use of the energy-based device, which was well described in the literature as leading to a reduction in operative times, was slightly greater, but with a statistically significant difference, in the HT Group [38,39].

Regarding our results on complications and operative times, we believe they are justified by the fact that, in these patients, a completion thyroidectomy is performed on a “virgin” surgical field, free of anatomical distortions resulting from a previous operation, which does not entail any particular technical difficulty for the surgeon. Actually, it is important to note that, in addition to the completion thyroidectomy, a central neck dissection was performed in 19.70% of the patients, making it necessary to return to the side of the first surgery. However, despite this, the complication rate in patients who underwent a completion thyroidectomy was comparable to that of patients submitted to primary thyroid surgery (both a total thyroidectomy and hemithyroidectomy).

Finally, about the timing of the completion thyroidectomy, the complication rate was comparable even though the completion surgery was, on average, performed 2.12 ± 1.20 months after the first operation, thus in the so-called “intermediate” postoperative period. For this reason, as claimed by other authors, we do not believe that the timing between the two surgical procedures has an influence on the development of complications [36].

As regards the rate of the completion thyroidectomy among patients undergoing a hemithyroidectomy, it was only 26.94%. Moreover, the completion surgery was not necessary in all patients submitted to a hemithyroidectomy with a subsequent histological finding of malignancy, but it was in 56.90%. Thus, following the ATA guidelines, unnecessary overtreatment, represented by total thyroidectomy, was avoided in a high percentage of patients. In this regard, it is also important to underline that a total thyroidectomy is burdened by a higher overall risk of complications, with specific reference to hypoparathyroidism and bilateral recurrent laryngeal nerve injury, which cannot occur in patients submitted to a hemithyroidectomy. Furthermore, as is well known and described by our data (reported in detail in Table 2), in patients undergoing a total thyroidectomy, the use of intraoperative nerve monitoring (useful for avoiding bilateral recurrent laryngeal nerve injury), the use of a drain, the operative time and the postoperative stay are greater. These elements together with the higher complication rate make the overall costs (both direct and indirect) of a total thyroidectomy higher than those of a hemithyroidectomy [7,12].

The main limitation of our study is that it is based on a retrospective analysis, and thus at risk of bias.

## 5. Conclusions

The risk of complications in patients undergoing a completion thyroidectomy after a hemithyroidectomy for a thyroid nodule with an indeterminate cytology was comparable to that of patients submitted to primary thyroid surgery (both a total thyroidectomy and hemithyroidectomy). Furthermore, also the result on operative times, obtained from the comparison between unilateral procedures (a hemithyroidectomy and a completion thyroidectomy), suggests that, in these patients, completion surgery does not entail any particular technical difficulty for the surgeon.

Based on our findings, we believe that, when possible, in accordance with the American Thyroid Association guidelines, in patients with thyroid nodular disease with an indeterminate cytology, a hemithyroidectomy should always be performed as the initial surgery. This surgical strategy allows to avoid unnecessary overtreatment and the complications and overall costs associated with a total thyroidectomy, without an increased risk of complications in the case of a need for a completion thyroidectomy.

Given the main limitation of our study, further prospective studies are needed to better investigate this topic.

## Figures and Tables

**Table 1 cancers-14-02472-t001:** Demographic and preoperative data regarding the whole sample, TT Group and HT Group.

	Total	TT Group	HT Group	*p* Value
	(n = 751)	(n = 506)	(n = 245)	
Sex				
- Male	242 (32.22%)	147 (29.05%)	95 (38.78%)	0.008
- Female	509 (67.78%)	359 (70.95%)	150 (61.22%)	
Age (years, mean ± SD)	51.74 ± 14.15	52.74 ± 13.72	49.67 ± 14.82	0.005
Familial history of DTC	73 (9.72%)	50 (9.88%)	23 (9.39%)	0.830
Hyperthyroidism	84 (11.19%)	67 (13.24%)	17 (6.94%)	0.010
US findings				
- Nodule size (mm, mean ± SD)	25.08 ± 13.21	25.10 ± 13.41	25.04 ± 12.82	0.957
- Bilateral nodules	334 (44.47%)	333 (65.81%)	1 (0.41%)	<0.001
Cytological category				
- TIR3A	262 (34.89%)	162 (32.02%)	100 (40.82%)	
- TIR3B	489 (65.11%)	344 (67.98%)	145 (59.18%)	0.018

SD: standard deviation; DTC: differentiated thyroid carcinoma; US: ultrasound.

**Table 2 cancers-14-02472-t002:** Information about surgical procedure, postoperative stay, complications and histological findings regarding the whole sample, TT Group and HT Group.

	Total	TT Group	HT Group	*p* Value
	(n = 751)	(n = 506)	(n = 245)	
Operative time (minutes, mean ± SD)	78.40 ± 33.94	89.26 ± 34.21	55.96 ± 19.16	<0.001
Use of IONM	487 (64.85%)	347 (68.58%)	140 (57.14%)	0.002
Use of EBD	751 (100%)	506 (100%)	245 (100%)	-
Use of drain	491 (65.38%)	418 (82.61%)	73 (29.80%)	<0.001
Postoperative stay (days, mean ± SD)	2.19 ± 1.03	2.42 ± 1.06	1.72 ± 0.76	<0.001
Hypoparathyroidism				
- Transient	141 (18.77%)	141 (27.87%)	0	<0.001
- Permanent	12 (1.60%)	12 (2.37%)	0	0.011
Cervical haematoma	5 (0.67%)	4 (0.79%)	1 (0.41%)	1.000
RLN injury				
- Unilateral	24 (3.20%)	17 (3.36%)	7 (2.86%)	0.714
- Bilateral	0	0	0	-
- Transient	18 (2.40%)	13 (2.57%)	5 (2.04%)	0.657
- Permanent	6 (0.80%)	4 (0.79%)	2 (0.82%)	1.000
Wound infection	1 (0.13%)	1 (0.20%)	0	1.000
Histological findings				
- Benign disease	390	261 (51.58%)	129 (52.65%)	0.783
- DTC	361	245 (48.42%)	116 (47.35%)	
- Microcarcinoma	133	102 (20.16%)	31 (12.65%)	0.012
- LN metastasis	20	11 (2.17%)	9 (3.67%)	0.231

SD: standard deviation; IONM: intraoperative nerve monitoring; EBD: energy-based device; RLN: recurrent laryngeal nerve; DTC: differentiated thyroid carcinoma; LN: lymph node.

**Table 3 cancers-14-02472-t003:** General information about CT Group.

	CT Group
	(n = 66)
% CT/HT	26.94%
% CT/HT with a histological finding of malignancy	56.90%
Two-stage thyroidectomy for LOS at IONM	1 (1.52%)
CND	13 (19.70%)
Interval between HT and CT (months, mean ± SD)	2.12 ± 1.20
Malignancy in the contralateral lobe	16 (24.24%)
LN metastasis	9 (13.64%)

CT: completion thyroidectomy; HT: hemithyroidectomy; LOS: loss of signal; IONM: intraoperative nerve monitoring; CND: central neck dissection; SD: standard deviation; LN: lymph node.

**Table 4 cancers-14-02472-t004:** Complications in TT Group and HT + CT Group.

	TT Group	HT + CT Group	*p* Value
	(n = 506)	(n = 66)	
Hypoparathyroidism			
- Transient	141 (27.87%)	14 (21.21%)	0.253
- Permanent	12 (2.37%)	2 (3.03%)	0.670
Cervical haematoma	4 (0.79%)	0	1.000
RLN injury			
- Unilateral	17 (3.36%)	4 (6.06%)	0.288
- Bilateral	0	0	-
- Transient	13 (2.57%)	3 (4.55%)	0.414
- Permanent	4 (0.79%)	1 (1.52%)	0.460
Wound infection	1 (0.20%)	0	1.000

RLN: recurrent laryngeal nerve.

**Table 5 cancers-14-02472-t005:** Data about surgical procedure, postoperative stay and complications in HT Group and CT Group.

	HT Group	CT Group	*p* Value
	(n = 245)	(n = 66)	
Operative time (minutes, median and IQR)	50 (42–65)	50 (40–60)	0.757
Use of IONM	140 (57.14%)	33 (50%)	0.300
Use of EBD	245 (100%)	64 (96.97%)	0.045
Use of drain	73 (29.80%)	19 (28.79%)	0.873
Postoperative stay (days, mean ± SD)	1.72 ± 0.76	1.77 ± 0.46	0.606
Cervical haematoma	1 (0.41%)	0	1.000
RLN injury			
- Unilateral	7 (2.86%)	2 (3.03%)	1.000
- Transient	5 (2.04%)	1 (1.52%)	1.000
- Permanent	2 (0.82%)	1 (1.52%)	0.512
Wound infection	0	0	-

IQR: interquartile range; IONM: intraoperative nerve monitoring; EBD: energy-based device; SD: standard deviation.

## Data Availability

The data that support the findings of this study are available from the corresponding author upon reasonable request.

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
