# Peer review of "Risk of Complications in Patients Undergoing Completion Thyroidectomy after Hemithyroidectomy for Thyroid Nodule with Indeterminate Cytology: An Italian Multicentre Retrospective Study"

_cancers, 2022, doi:10.3390/cancers14102472_

Round 1

Reviewer 1 Report

The paper is of interest and discuss current debate. Outlines are clear.

Minor comments

Methodology:

Suggest formating text into paragraphs rather than bullets.

Genomic data was collected? final pathology TNM state?

Results:

Table 5. Operative time is non-parametric thus should be presented as median (IQR) and corresponding MW test should be used.

Adding data on risk of malignancy in these indeterminant nodules and false positive/negative rates will be useful.

Also, suggest multivariate regression analysis to predict complications (overall) including patient characteristics, imaging, type of surgery, ....etc

Author Response

Dear Reviewer 1,

thank you for your careful analysis of our manuscript and for your constructive and precious suggestions.

Below, the answers, point by point, to your comments.

1) As you suggested, we formatted the "Materials and Methods" section into paragraphs rather than bullets.

2) As the aim of this study was to assess the complication rate rather than the DTC behaviour, we did not collect these data.

3) As you correctly suggested, operative time (in Table 5 and the corresponding text) was presented as median (IQR) and analysed through Mann-Whitney U test.

4) Data on the malignancy rate at final histological examination are reported in Table 2.

Regarding the false negative rate, it cannot be assessed by our analysis as patients with benign cytology (TIR2 or Bethesda II) were not included in this study.

5) We think this suggestion is interesting, however, the aim of this study was to assess the occurrence of complications depending on the type of surgical approach (total thyroidectomy vs hemithyroidectomy + completion thyroidectomy), regardless of patient and disease characteristics. As no statistically significant difference was found in terms of complications, we believe it is superfluous to evaluate the influence of other factors.

Best regards.

Reviewer 2 Report

The authors are studying an important topic which was to assess the rate of complications in patients undergoing completion thyroidectomy after hemithyroidectomy for thyroid nodule with indeterminate cytology.

Comments:

  1. In the introduction section “In about 25% of cases FNAC leads to the diagnosis of indeterminate thyroid nodule, which represents a problem, since malignancy, although  relatively low (up to 30%), cannot be excluded with certainty. For this reason, patients with this diagnosis undergo surgery with a diagnostic rather than therapeutic purpose, with almost 80% of surgical procedures unnecessary”

---Should also include that diagnostic surgery is not the only option, as per the ATA guidelines molecular testing is also an option; per ATA “For nodules with AUS/FLUS cytology, after consideration of worrisome clinical and sonographic features, investigations such as repeat FNA or molecular testing may be used to supplement malignancy risk assessment in lieu of proceeding directly with a strategy of either surveillance or diagnostic surgery”

  1. Study design – specify indeterminate cytology, does this include Bethesda III and IV?
  2. Although not the main purpose of the study will be interesting to see the complication rate of those patients with underlying hyperthyroidism.

Author Response

Dear Reviewer 2,

thank you for your careful analysis of our manuscript and for your constructive and precious suggestions.

Below, the answers, point by point, to your comments.

1) As you correctly suggested, we specified that diagnostic surgery is not the only option in patients with thyroid nodular disease with indeterminate cytology.

2) As specified in the "Materials and Methods" section, “the results of fine-needle aspiration cytology were classified according to the 2014 Italian consensus of the Italian Thyroid Association (AIT), the Italian Association of Clinical Endocrinologists (AME), the Italian Society of Endocrinology (SIE) and the Italian Society for Anatomic Pathology and Cytology joint with the Italian Division of the International Academy of Pathology (SIAPEC-IAP) for the classification and reporting of thyroid cytology”.

TIR3A and TIR3B categories of this classification correspond to Bethesda III and IV.

The stratification of patients according to these categories is shown in Table 1.

3) We think this suggestion is interesting, however, as you also state, it is not an aim of this study. In addition, the risk of complications is mainly increased in the case of Graves' disease for which the surgical indication is total thyroidectomy, thus hemithyroidectomies and completion thyroidectomies would be excluded from the analysis.

Best regards.